

# The feasibility of the posterior tibial nerve-flexor hallucis brevis pathway applied in neuromuscular monitoring: a multicentric, controlled, and prospective clinical trial

Weiqiang Chen[1,*], Zhijian Chen[1,*], Fang Cheng[2], Zhuodan Wang[3], Juan Li[4], Shangrong Li[1] and Hanbin Xie[1]

[1] Department of Anesthesiology, The Third Affiliated Hospital of Sun Yat-sen University, Guangzhou, China
[2] Department of Anesthesiology, Jiangmen Central Hospital, Jiangmen, China
[3] Department of Anesthesiology, The Second Affiliated Hospital of Guangzhou Medical University, Guangzhou, China
[4] GCP ClinPlus Co., Ltd, Guangzhou, China
* These authors contributed equally to this work.

Corresponding authors
Shangrong Li,
lishangr@mail.sysu.edu.cn
Hanbin Xie,
xiehanb@mail.sysu.edu.cn

## ABSTRACT

This study aimed to investigate the clinical viability of utilizing the flexor hallucis brevis as an alternative site for neuromuscular monitoring compared to the conventional adductor pollicis. Patients were recruited from three medical centers. Cis-atracurium was administered, and two monitors were employed independently to assess neuromuscular blockade of the adductor pollicis and the ipsilateral flexor hallucis brevis, following a train of four (TOF) pattern until TOF ratios exceeded 0.9 or until the conclusion of surgery. Statistical analysis revealed significant differences in onset time, duration of no-twitch response, spontaneous recovery time, and total monitoring time between the two sites, with mean differences of −53.54 s, −2.49, 3.22, and 5.89 min, respectively ($P < 0.001$).The posterior tibial nerve-flexor hallucis brevis pathway presents a promising alternative for neuromuscular monitoring during anesthesia maintenance. Further investigation is warranted to explore its utility in anesthesia induction and recovery.

**Trial registration:** The trial was registered at www.chictr.org.cn (20/11/2018, ChiCTR1800019651).

## INTRODUCTION

Residual neuromuscular blockade (RNMB) has emerged as a prevalent complication of general anesthesia following the introduction of neuromuscular blocking agents (NMBAs) (*Thilen et al., 2023*). A retrospective analysis in 2020 (*Raval et al., 2020*) revealed a median RNMB incidence of 30% upon arrival at the post-anesthesia care unit (PACU). RNMB

compromises the function of upper airway muscles (*Eriksson et al., 1997*; *Sundman et al., 2000*) and the hypoxic ventilatory response (HVR) (*Eriksson, Sato & Severinghaus, 1993*), predisposing patients to heightened risks of respiratory complications (*Asai & Isono, 2014*) and prolonged intubation and PACU stays (*Murphy et al., 2004*, *2003*). Perioperative quantitative neuromuscular monitoring (*Gätke et al., 2002*; *Murphy et al., 2011*; *Todd & Hindman, 2015*) has been internationally endorsed to mitigate RNMB by evaluating the degree of neuromuscular blockade and guiding the administration of NMBAs and their antagonists. However, the clinical implementation of quantitative neuromuscular monitoring remains limited (*Murphy & Brull, 2022*; *Naguib et al., 2010*; *Weigel et al., 2022*).

The prevailing approach to quantitative neuromuscular monitoring involves stimulating the ulnar nerve and observing the resultant movement of the adductor pollicis *via* acceleromyography (AMG) (*Murphy, 2018*). When monitored by AMG, the movement of muscle should be completely free of any external force. However, limitations exist in using the adductor pollicis as a monitoring site. Firstly, venous access and various anesthetic monitoring devices (such as invasive blood pressure monitor and pulse oximetry) are typically situated on the hands, potentially impeding the movement of the adductor pollicis. Secondly, achieving optimal positioning to ensure unhindered movement of the thumb, typically requiring arm extension, is frequently challenging during quantitative neuromuscular monitoring, as reported in a survey, due to its potential interference with surgeons and surgical equipment (*Söderström et al., 2017*). Consequently, there arises a need for alternative monitoring sites to enhance the accessibility and utility of quantitative neuromuscular monitoring. The challenges associated with using the adductor pollicis for neuromuscular monitoring underscore the need for an alternative site.

Stimulating the posterior tibial nerve induces contraction of the flexor hallucis brevis, akin to the adductor pollicis (*Sopher, Sears & Walts, 1988*), with fewer potential sources of interference, making it a promising alternative site. Previously, several studies have compared the time course between the adductor pollicis and the flexor hallucis brevis (*Betz et al., 2020*, *2021*; *Heier & Hetland, 1999*; *Kern et al., 1997*; *Kitajima, Ishii & Ogata, 1996*; *Saitoh et al., 1998*). The posterior tibial nerve stimulation may alleviate surgical site compression and circumvent potential interference from intravenous access and specific anesthesia monitoring devices. However, these studies had relatively small sizes, with many (*Heier & Hetland, 1999*; *Kern et al., 1997*; *Saitoh et al., 1998*) utilizing inhalation anesthesia, which is known to influence NMBAs. Furthermore, some studies (*Heier & Hetland, 1999*) employed different methods to detect muscle contraction, rendering the collected data incomparable. Consequently, the clinical utility of the flexor hallucis brevis for neuromuscular monitoring remains uncertain.

Therefore, the primary objective of this study was to compare the onset and spontaneous recovery of neuromuscular blockade between the adductor pollicis and the flexor hallucis brevis muscles. Furthermore, the clinical feasibility of utilizing the flexor hallucis brevis as an alternative site for perioperative neuromuscular monitoring was investigated.

## METHODS

This multicentric, prospective, open, controlled, and observational clinical trial was conducted at the Third Affiliated Hospital of Sun Yat-Sen University (Guangzhou, Guangdong), the Second Affiliated Hospital of Guangzhou Medical University (Guangzhou, Guangdong), and Jiangmen Central Hospital (Jiangmen, Guangdong). The trial was registered at the Chinese Clinical Trial Registry (20/11/2018, ChiCTR1800019651). The primary objective was to assess the feasibility of using the posterior tibial nerve-flexor hallucis brevis pathway for quantitative neuromuscular monitoring, using the ulnar nerve-adductor pollicis pathway as a reference standard.

### Patients

Patients scheduled for elective surgery under general anesthesia with endotracheal intubation were recruited from the three centers. Inclusion criteria were as follows: (1) aged 18–60 years; (2) American Society of Anesthesiologists (ASA) classification grade of I or II; (3) a body mass index (BMI) between 18 and 30 kg/m$^2$; (4) no personal or family history of neuromuscular junction diseases.

Exclusion criteria included: (1) history of nerve injury or use of drugs affecting the neuromuscular transmission or both; (2) dermatitis, ulcers, or other lesions at the monitored sites; (3) history of certain medical conditions such as diabetes, stroke, cerebral hemorrhage, and intervertebral disc disorder; (4) severe cardiac, pulmonary, hepatic, or renal dysfunction; (5) those undergoing microsurgery or surgery necessitating unique body positions.

Participants were considered to have dropped out if any of the following situations occurred: (1) the patient requested withdrawal of consent; (2) initial calibration failed twice or more; (3) any monitored site was found to be compressed twice or more; (4) additional NMBAs were administered before the end of data collection; (5) T1 (train-of-four count (TOFC) = 1) did not recover before the conclusion of surgery; (6) any severe adverse event occurred during monitoring.

The study was approved by the Institutional Review Board of the Third Affiliated Hospital of Sun Yat-sen University (IRB (2018) 02-348-01). All patients were briefed on the experiment's purpose, details, and potential risks, and provided informed consent before inclusion. Each patient was assigned an ID number.

### Intraoperative monitoring

Before induction, conventional monitoring was conducted, including noninvasive blood pressure (NIBP), pulse oximetry saturation (SpO$_2$), and electrocardiography (ECG). Ventilator monitoring of airway pressure and partial pressure of end-tidal carbon dioxide (P$_{ET}$CO$_2$) commenced post-intubation.

NMB was monitored using the neuromuscular transmission module of BeneVision N12 Patient Monitor (Shenzhen Mindray Bio-Medical Electronics Co., Ltd, Guangdong, China). The upper limb for monitoring was abducted and secured with the palm facing upward. Two electrodes were positioned on the ulnar side of the wrist along the ulnar nerve pathway, spaced 2–3 cm apart. An acceleration sensor was affixed to the palmar side

of the thumb. Similarly, the ipsilateral lower limb was positioned naturally on the operating table, with electrodes attached to the medial malleolus along the posterior tibial nerve pathway. The acceleration sensor was secured to the plantar side of the big toe. U-shaped barriers shield the monitored thumb and toe from external force interference, with sterile towels covering these barriers to maintain the monitored site temperature above 32.5 °C (Figs. S1 and S2).

## Anesthetic protocol

At induction, midazolam 0.05–0.10 mg/kg, propofol 2.0–2.5 mg/kg, and fentanyl 2–4 µg/kg were administered for sedation and analgesia. Once consciousness was achieved, two AMG modules were calibrated synchronously to establish a suitable electric current and baseline movement. Upon successful calibration, cis-atracurium 0.2 mg/kg was administered intravenously. Stimulation and evaluation were carried out simultaneously by the two modules every 12 s. Intubation was performed once TOFC reached 0 at the hand. During surgery, anesthesia was maintained using propofol and remifentanil *via* target-controlled infusion (TCI). Propofol was initiated at a plasma concentration of 4–6 µg/mL and remifentanil at 4–8 ng/mL Anesthetic adjustments were made based on intraoperative hemodynamic changes, with propofol and remifentanil adjusted by ±0.5 µg/mL or ±0.5 ng/mL, respectively, to maintain NIBP fluctuations within ±30% of baseline.

Monitoring was terminated at the end of surgery or when the TOF ratio at both sites reached 0.9 after the first dose of cis-atracurium. Additional NMBAs could be administered if necessary to optimize surgical conditions.

## Data collection and endpoints

Researchers recorded patients' general information and the timing of induction, calibration, cis-atracurium administration, intubation, and cessation of NMB monitoring. The BeneVision N12 Patient Monitor automatically documented the TOF values and their corresponding times at both monitored sites.

The primary endpoints included: (1) onset time (OT), defined as the duration from cis-atracurium administration to the point where TOFC faded to 0; (2) period of no-twitch response (NTR), representing the interval from cis-atracurium administration to recovery of T1 (TOFC = 1); (3) spontaneous recovery time (SRT), denoting the duration from cis-atracurium administration to TOFR reaching 0.5; (4) total time (TT), indicating the duration from cis-atracurium administration to TOFR reaching 0.9.

## Statistical analysis

Based on the report by *Heier & Hetland (1999)*, it was expected that over 95% of the onset time difference between the thumb and toe would fall within −21 to −113 s, and the spontaneous recovery time difference would fall within −7 to −1 min. With a significance level (α) of 0.01, a test power (1-β) of 90%, and a dropout rate of 20%, the sample size was calculated to be 200 patients based on single-arm objective performance criteria.
Continuous data were presented as mean ± standard deviation (SD), while categorical data were expressed as numbers and percentages (%). Student's paired t-test was used to compare means between thumb and toe monitoring results, and the mean difference between thumb and toe, along with its 95% CI, was reported using standard error. One-way ANOVA and Fisher's LSD test for *post-hoc* comparison were employed to compare mean differences among the three centers (S/G/J). Two-way mixed-design ANOVA was utilized to investigate the main and interaction effects between sites and centers. Pearson's correlation coefficient was used for two continuous variables, while Spearman's rho was used for categorical variables. The difference between the two percentages was assessed using the z-score and z-test. A two-tailed significance level of $P < 0.05$, was applied to all tests. All analyses were conducted using IBM SPSS Version 25 (SPSS Statistics V25, IBM Corporation, Somers, New York, USA); GraphPad Prism (version 8.2.1) was used to generate the Bland–Altman plots illustrating monitoring results between the thumb and toe.

## RESULTS

### Patients' general information

Between March 2019 and January 2020, 213 patients met the inclusion criteria. As depicted in Fig. 1, 154 patients were included and analyzed in this study. Table 1 presents patients' mean age and BMI as 43.70 ± 15.43 years and 22.92 ± 3.32 kg/m$^2$, respectively, with a gender ratio of 1:1.33 (men/women = 66/88). Among the patients, 109 (70.78%) had ASA grade I. The predominant surgical types were general surgery (44.80%) and otolaryngology (44.80%), with gynecology (7.79%) and orthopedics (2.59%) comprising the remaining cases. The 154 patients were sourced from three different centers, including the Third Affiliated Hospital of Sun Yat-Sen University (Center S, $n = 73$), the Second Affiliated Hospital of Guangzhou Medical University (Center G, $n = 38$), and the Jiangmen Central Hospital (Center J, $n = 43$) (Fig. 2).

### Monitoring time intervals between thumb and toe

Figure 3 shows the proportion of hands faster or slower than feet during a given period, including OT, NTR, SRT, and TT. Table 2 presents the monitoring time intervals between the thumb and toe, encompassing OT, NTR, SRT, and TT. Furthermore, the mean difference (thumb minus toe) and its 95% confidence interval [CI] based on standard error are reported. All time intervals significantly differed between the thumb and toe (all $P < 0.001$). Specifically, the OT and NTR durations were significantly longer at the foot, while the SRT and TT durations were prolonged at the hand.

### Comparisons of mean differences among three centers

Table 3 presents the mean differences in monitoring results across the three centers. The ANOVA results and *post-hoc* comparisons between paired centers did not yield significant differences (all $P > 0.05$), indicating comparability among the three centers.

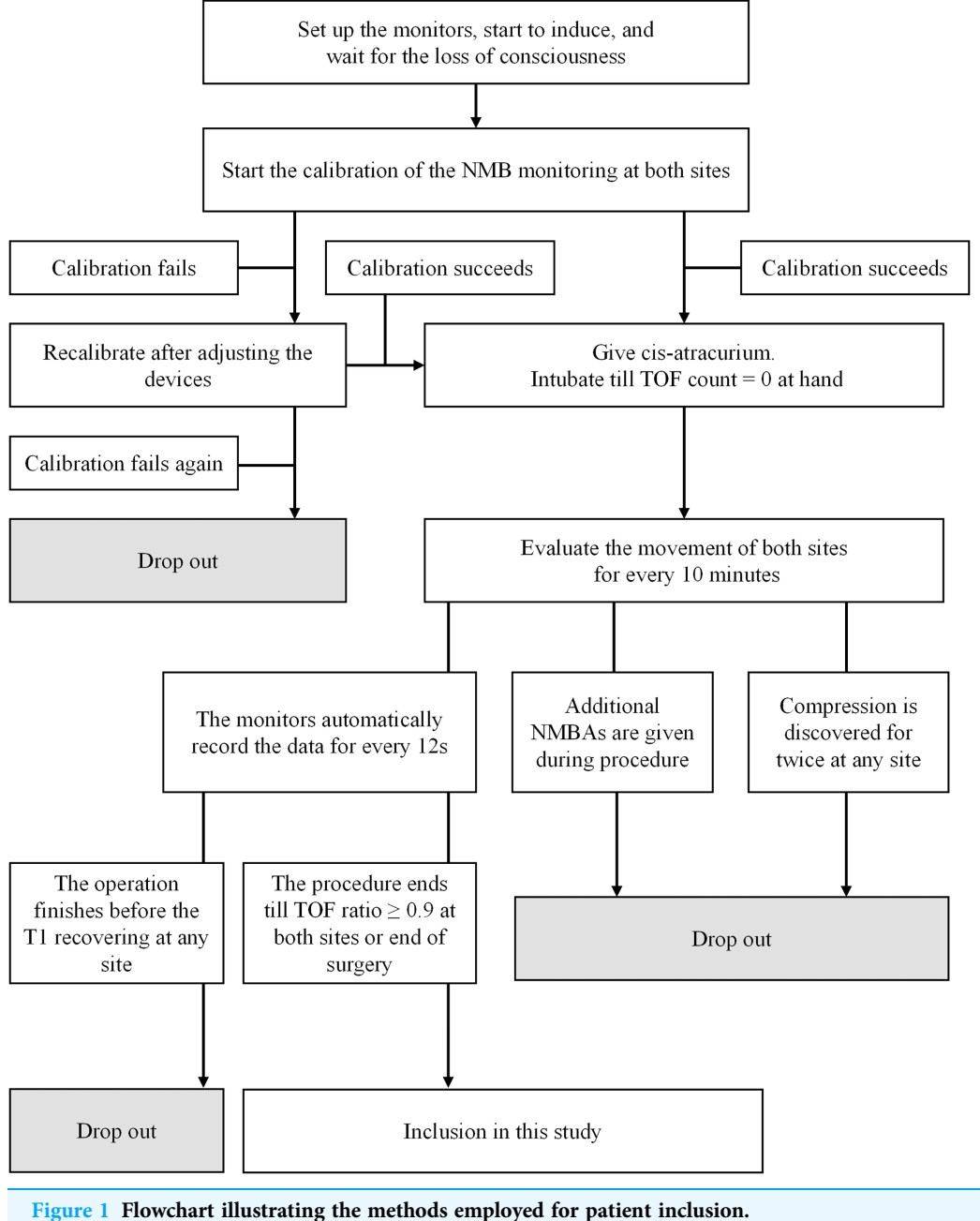

**Figure 1 Flowchart illustrating the methods employed for patient inclusion.**

## Two-way ANOVA results

The results of two-way mixed-design ANOVA examining the main and interaction effects are indicated in Table 4. Notably, no interaction effect was observed (all $P > 0.05$). Although NTR and TT results differed significantly among centers, the mean difference was not significant among centers, as previously reported. The highly substantial $P$-values indicate the differences between the thumb and toe in all monitoring results (all $P < 0.001$).

| Table 1 Demographic characteristics of the patients. | |
|---|---|
| **Parameters** | **Mean ± SD, N (%)** |
| Age, year | 43.70 ± 15.43 |
| Sex | |
| Male | 66 (42.86%) |
| Female | 88 (57.14%) |
| Height, cm | 163.58 ± 7.76 |
| Weight, kg | 61.48 ± 10.67 |
| BMI, kg/m$^2$ | 22.92 ± 3.32 |
| ASA | |
| I | 109 (70.78%) |
| II | 45 (29.22%) |
| Source | |
| Center S | 78 (50.65%) |
| Center G | 35 (22.73%) |
| Center J | 41 (26.62%) |
| Surgical type | |
| General surgery | 71 (46.10%) |
| Otalaryngology | 66 (42.86%) |
| Gynecology | 12 (7.79%) |
| Orthopedics | 3 (1.95%) |
| Urology | 2 (1.30%) |

## Bland–alman plots

Figure 4 depicts the Bland–Alman plots of all four monitoring results. While no specific pattern was observed in these differences, an overall trend of negative results for OT and NTR, and positive results for SRT and TT was observed. The percentages of positive/ negative difference counts in monitoring results (Fig. 3) further corroborate the trends identified in Bland–Alman plots. Specifically, significantly higher rates of 'Thumb ≤ Toe' were observed in OT and NTR, while higher rates of 'Thumb > Toe' were found in SRT and TT (all $P < 0.001$).

## Association between demographics and monitoring results

Table 5 presents the findings of the correlation coefficient analysis between monitoring results (OT, NTR, SRT, and TT) and patient demographics (age, height, weight, BMI, gender, ASA grade, and surgical type). No significant associations were found (all $P > 0.05$).

## DISCUSSION

As an exploratory experiment, this study elucidated the regularities of neuromuscular transmission under total intravenous anesthesia (TIVA), monitored by AMG at the flexor hallucis brevis or adductor pollicis. Furthermore, the clinical feasibility of using the flexor

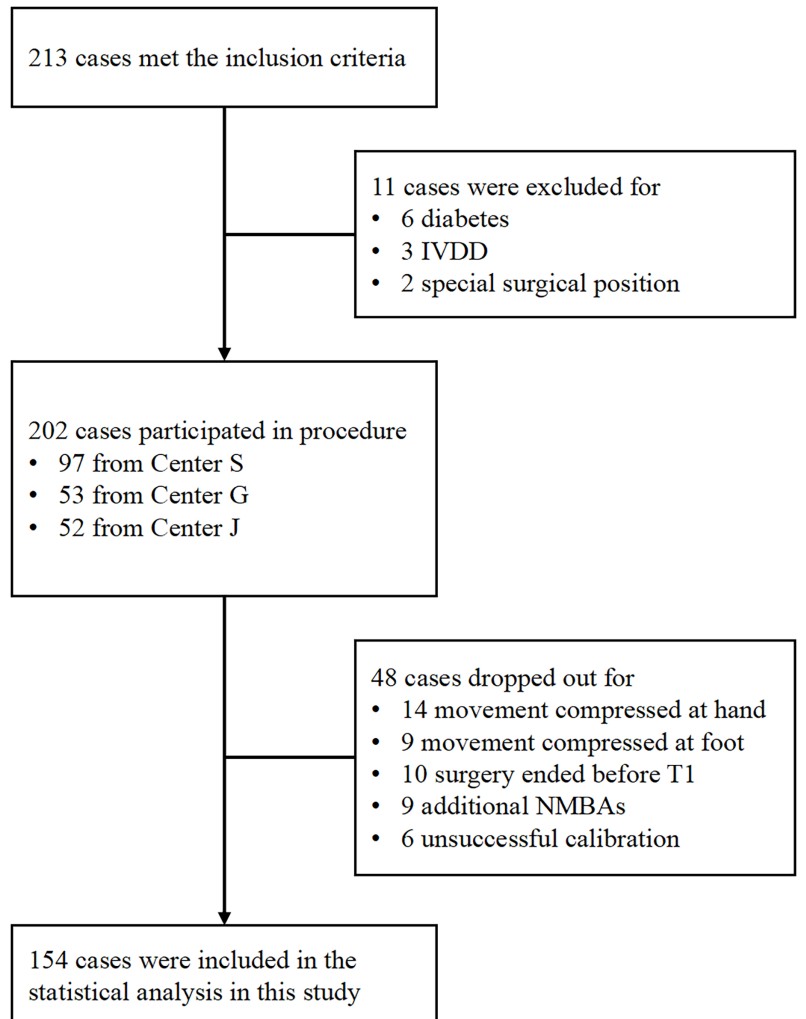

**Figure 2 Flowchart depicting the process of patient inclusion.**

hallucis brevis as an alternative site for neuromuscular blockage monitoring was investigated.

Following cis-atracurium administration, onset time was generally longer at the flexor hallucis brevis than at the adductor pollicis. This finding aligns with previous studies (*Heier & Hetland, 1999*; *Kern et al., 1997*; *Le Merrer et al., 2020*), potentially attributable to differences in blood flow distribution. The shorter distance from the heart pump results in higher blood flow per gram of muscle in the hand compared to the foot, leading to a more rapid increase in plasma drug concentration of NMBAs. However, this onset difference between the hand and foot is not observed in infants (*Kitajima, Ishii & Ogata, 1996*), whose blood distribution is relatively equal. In a previous study (*Le Merrer et al., 2020*), recovery of the first TOF twitch was significantly slower at flexor hallucis brevis compared with the adductor pollicis, which differs from our findings. The discrepancy in results is mainly attributed to the difference in sample size. Clinically, the absence of TOFC at the

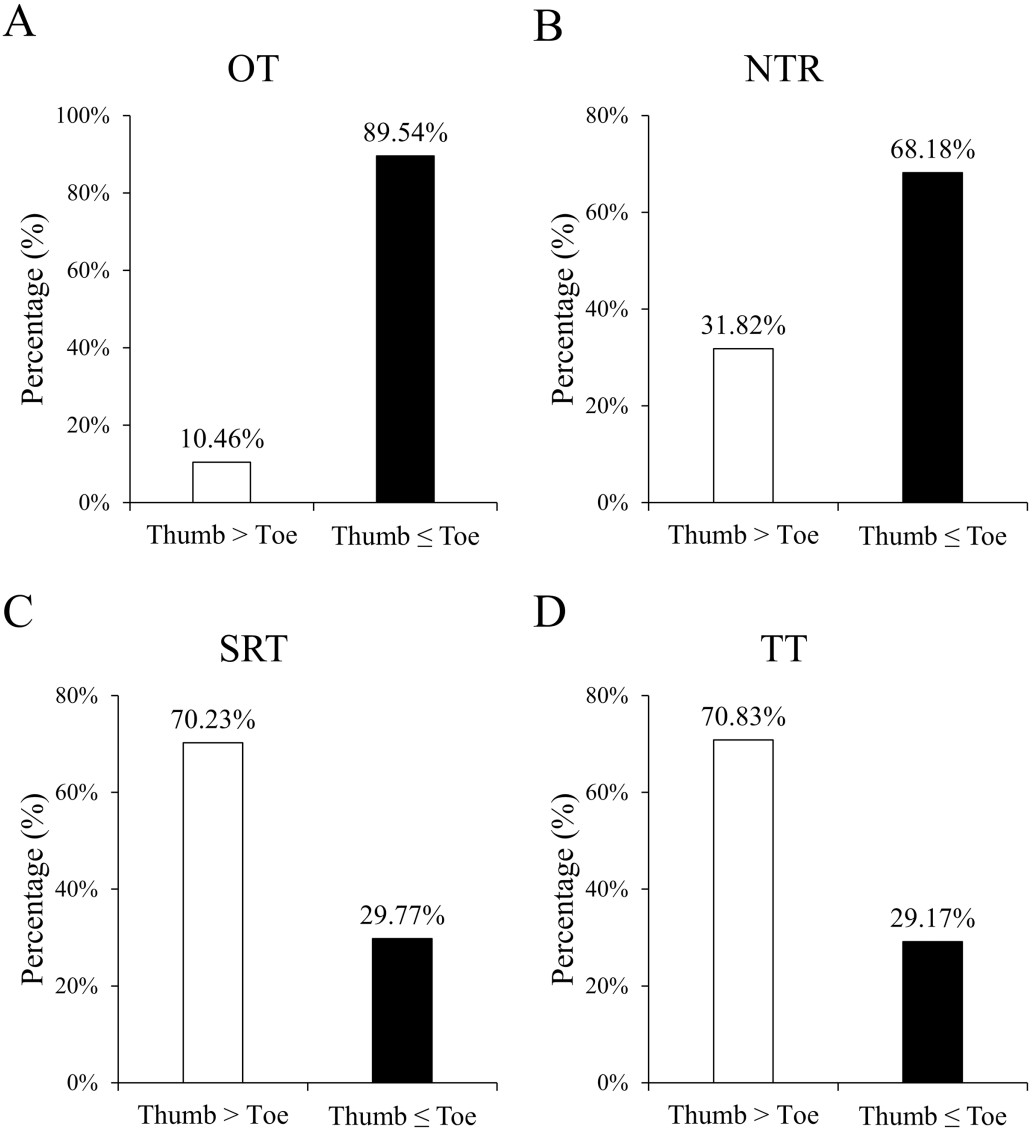

**Figure 3 Percentages of the counts of positive and negative differences in monitoring results between the thumb and toe, covering OT (A), NTR (B), SRT (C), and TT (D).** Positive difference indicates a longer duration for the thumb than the toe, whereas negative difference denotes an equal or shorter duration. All the percentages between the thumb and toe were significant (all $P < 0.001$).

hand is a recommended timing for tracheal intubation to reduce the incidence of vocal cord damage (*Lundstrøm et al., 2017*). However, when TOFC reached 0 at the hand, the mean TOFR was 0.68 at the foot; the Bland–Altman plot suggests significant individual variability among the onset-time difference. *Le Merrer et al. (2020)* suggest that flexor hallucis brevis could be an appealing alternative site for NMB monitoring when the adductor pollicis is inaccessible. Therefore, it is not recommended to adopt the flexor hallucis brevis as a site for neuromuscular monitoring during intubation.

On the other hand, theoretically, TOFR at the foot should recover more rapidly than that at the hand because the percentage of type II muscle fiber is higher in the flexor

**Table 2 Comparative analysis of monitoring results between the thumb and toe.**

| Period | Mean±SD | | Difference (hand minus foot) | | |
| | At hand | At foot | Mean ± SE (95% CI) | t | P |
|---|---|---|---|---|---|
| OT (s) | 232.01 ± 65.30 | 285.55 ± 71.87 | −53.54 ± 4.66 (−62.76 to −44.33) | −11.48 | <0.001 |
| NTR (min) | 56.21 ± 9.56 | 58.70 ± 9.92 | −2.49 ± 0.49 (−3.45 to −1.53) | −5.12 | <0.001 |
| SRT (min) | 14.93 ± 5.54 | 11.72 ± 5.38 | 3.22 ± 0.58 (2.07 to 4.37) | 5.53 | <0.001 |
| TT (min) | 94.91 ± 14.60 | 89.02 ± 14.84 | 5.89 ± 0.81 (4.28 to 7.49) | 7.27 | <0.001 |

Note:
Student's paired t-test was used and the 95% CI of mean difference was calculated using standard error. SE, standard error.

**Table 3 Differences between hand and foot monitoring results and their comparisons across the three centers.**

| Period | Difference (hand minus foot) | | | ANOVA | | P-value of post-hoc comparisons | | |
| | Center S | Center G | Center J | F | P | S vs. G | S vs. J | G vs. J |
|---|---|---|---|---|---|---|---|---|
| OT (s) | −43.15 ± 52.25 | −63.94 ± 63.13 | −64.68 ± 60.61 | 2.64 | 0.075 | 0.078 | 0.052 | 0.955 |
| NTR (min) | −1.70 ± 6.01 | −4.04 ± 6.23 | −2.67 ± 5.76 | 1.86 | 0.159 | 0.057 | 0.404 | 0.322 |
| SRT (min) | 3.33 ± 7.20 | 2.38 ± 6.73 | 3.83 ± 5.33 | 0.39 | 0.678 | 0.508 | 0.732 | 0.392 |
| TT (min) | 6.52 ± 10.26 | 6.51 ± 8.27 | 4.07 ± 5.82 | 0.88 | 0.419 | 0.993 | 0.211 | 0.299 |

Note:
One-way ANOVA and Fisher's LSD test as post-hoc comparison was used.

**Table 4 Results of two-way mixed-design ANOVA for monitoring results between the thumb and toe across the three centers.**

| Period | Site (hand/foot) | | Center (S/G/J) | | Interaction (Site × Center) | |
| | F | P | F | P | F | P |
|---|---|---|---|---|---|---|
| OT (s) | 135.96 | <0.001 | 0.55 | 0.577 | 2.64 | 0.075 |
| NTR (min) | 29.89 | <0.001 | 5.26 | 0.006 | 1.86 | 0.159 |
| SRT (min) | 26.02 | <0.001 | 1.58 | 0.211 | 0.39 | 0.678 |
| TT (min) | 43.46 | <0.001 | 6.82 | 0.002 | 0.88 | 0.419 |

Note:
Two main effects (site and center) and one interaction effect (site × center) were reported with F-value and P-value.

hallucis brevis, which is more resistant to NMBAs. However, a recent study shows a converse result in the initial stage of recovery (*Le Merrer et al., 2020*). Similarly, in our study, the mean period of no-twitch response is 56.21 min at the adductor pollicis compared to 58.70 min at the flexor hallucis brevis. In clinical practice, maintaining neuromuscular blockage at approximately TOFC at 1 or 2 is the recommended timing for additional NMBAs, possibly when T1 emerges. While monitoring the flexor hallucis brevis, the period of no-twitch response is 2.49 min longer than that at the adductor pollicis, which is clinically acceptable. Therefore, it is feasible to employ the timing when TOFC = 1 at the flexor hallucis brevis for additional NMBAs. As for the ideal timing for NMBA reversal, previous studies (*Fuchs-Buder et al., 2013*, *2010*; *Schaller et al., 2010*) indicate that when the TOF ratio recovers to 0.50 at the adductor pollicis, a low dose

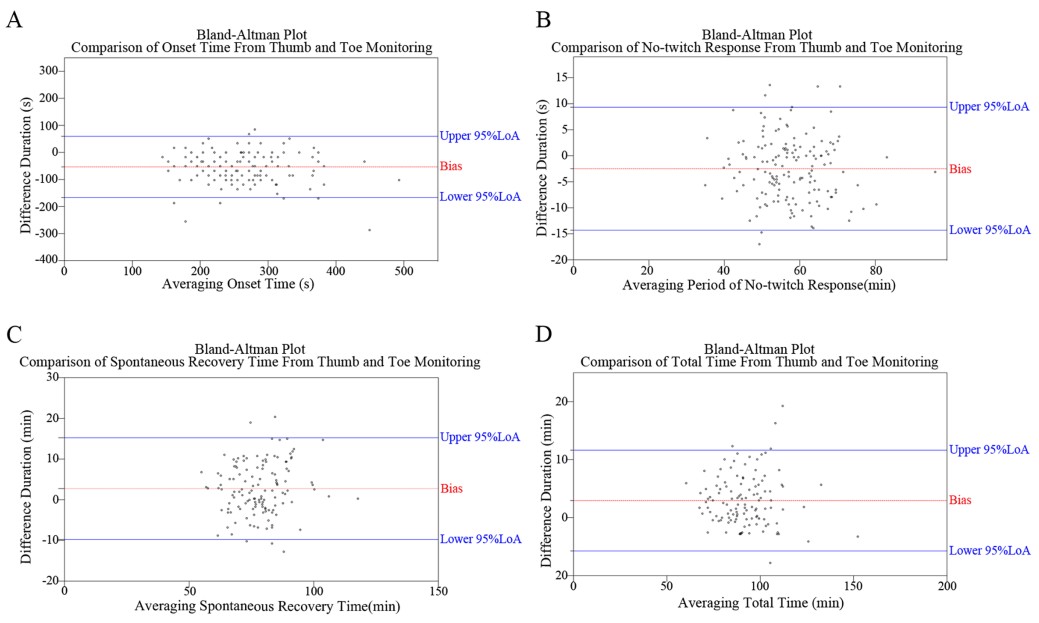

**Figure 4** Bland–Altman plots illustrating the comparative analysis of monitoring outcomes between thumb and toe, including assessments of onset time (OT) (A), no-twitching response (NTR) (B), spontaneous recovery time (SRT) (C), and total time (TT) (D).

**Table 5 Correlation coefficient analysis between monitoring results and patient demographics.**

| | Pearson's r | | | | Spearman's rho | | |
|---|---|---|---|---|---|---|---|
| Period | Age, year | Height, cm | Weight, kg | BMI, kg/m$^2$ | Gender | ASA grade | Surgical type |
| OT (s) | 0.078 | 0.148 | 0.139 | 0.092 | −0.045 | 0.384 | 0.014 |
| NTR (min) | −0.145 | −0.012 | −0.102 | −0.101 | 0.100 | 0.205 | 0.150 |
| SRT (min) | 0.105 | 0.089 | 0.086 | 0.051 | 0.018 | 0.102 | 0.022 |
| TT (min) | 0.051 | 0.017 | 0.031 | 0.043 | −0.026 | 0.605 | 0.086 |

**Note:**
Pearson's correlation coefficient was used between two continuous variables while Spearman's rho was used if one or both variables were categorical. No significant correlation coefficient ($P < 0.05$) was found in these results.

(0.02–0.03 mg/kg) of neostigmine is adequate to antagonize the muscle blockade within a short time completely. Our results show that the spontaneous recovery time of the flexor hallucis brevis is 3.22 min faster than that of the adductor pollicis. Moreover, the mean TOFR is 0.60 at the foot and 0.50 at the hand. Therefore, it is advisable to administer neostigmine when the TOFR ratio reaches 0.60 if the posterior tibial nerve-flexor hallucis brevis is adopted; however, caution should be exercised due to significant individual variation, and monitoring should continue until complete recovery of neuromuscular transmission.

In most cases, a TOFR ≥ 0.90 at the adductor pollicis indicates complete recovery of respiratory muscle blockade and an appropriate timing for extubation. However, when monitored by AMG, a TOFR ≥ 1.0 has been considered a better indicator of complete recovery from neuromuscular blockage (*Ungureanu et al., 1993*). It has recently been

reported that the TOFR usually exceeds 1.0 after calibration, complicating subsequent interpretation (*Schreiber, Mucha & Fuchs-Buder, 2011*). This finding suggests that more time should be allowed when AMG monitors to ensure safe extubation even if the TOFR reaches 0.9 at hand. Since our data demonstrates that total time at the foot is significantly longer than at the hand, further study is warranted to explore the cut-off value for extubation of the flexor hallucis brevis as a monitoring site during anesthesia recovery.

Bland–Altman analysis aims to evaluate the consistency between two measurement methods, reflecting whether they can be used interchangeably. Consistency evaluation relies on the number of data points outside the 95% concordance limit, the maximum difference within the concordance limit, and clinical acceptability. This study's significant individual variations between the two pathways, observed from the Bland–Altman plots and the corresponding distribution of TOF ratios, should not be overlooked. Furthermore, correlational analyses suggest no significant influence of demographic characteristics. Several hypothetical explanations for the observed variation between the pathways. Firstly, muscle fiber size influences reactivity to NMBAs (*Ibebunjo, Srikant & Donati, 1996*), with smaller fibers leading to neuromuscular transmission recovery. Variations in muscle group sizes between the hand and foot may contribute to this disparity. Secondly, intraoperative adjustments in the supine position, such as Trendelenburg or reverse Trendelenburg, affect blood flow differently in the upper and lower limbs, potentially influencing neuromuscular transmission. Thirdly, idiosyncrasies in AMG also play a role. While mechanomyography (MMG) is the gold standard for neuromuscular transmission detection, it is not routinely used clinically due to its bulky shape and complex setup process. In contrast, the 95% limits of agreement (LoA) of AMG used in this study were (−19%, +24%) when the TOF ratio = 0.90 (*Dubois et al., 2014*). Further investigation is warranted to understand the reasons for the variation between the two pathways, emphasizing the importance of neuromuscular transmission monitoring.

This study has several limitations. Firstly, while the adductor pollicis is commonly used in clinical practice, its drug response may differ from that of laryngeal adductor muscles (*Capron et al., 2004*). Comparing the flexor hallucis brevis to the laryngeal adductor would provide a more precise assessment of feasibility and safety as a monitoring site. However, clinical monitoring of NMB at laryngeal adductor muscles *via* AMG has not been achieved. Secondly, only TIVA was utilized in this study. Since the impact of inhaled anesthetics on neuromuscular blockade has been reported (*Ye et al., 2015*), different types of general anesthesia should be considered. Thirdly, the study only explored neuromuscular recovery after the initial dose of cis-atracurium without additional or antagonistic interventions. Therefore, considering these limitations, further research should explore flexor hallucis brevis as a monitoring site under different anesthesia methods, various muscle relaxants, or different monitoring sites.

## CONCLUSIONS

In summary, the posterior tibial nerve-flexor hallucis brevis pathway shows promise as a feasible alternative in neuromuscular transmission monitoring during anesthesia

maintenance. Further investigation is needed to understand its role in anesthesia induction and recovery.

## ACKNOWLEDGEMENTS

We thank Shenzhen Mindray Bio-Medical Electronics Co., Ltd. (Nanshan, Shenzhen, P. R. China) for providing the neuromuscular transmission devices (BeneVision N12 Patient Monitor).

### Funding
The authors received no funding for this work.

### Competing Interests
Juan Li is employed by GCP ClinPlus Co., Ltd. The authors declare that they have no competing interests.

### Author Contributions
- Weiqiang Chen conceived and designed the experiments, performed the experiments, prepared figures and/or tables, authored or reviewed drafts of the article, and approved the final draft.
- Zhijian Chen performed the experiments, analyzed the data, authored or reviewed drafts of the article, and approved the final draft.
- Fang Cheng conceived and designed the experiments, performed the experiments, analyzed the data, prepared figures and/or tables, and approved the final draft.
- Zhuodan Wang conceived and designed the experiments, performed the experiments, prepared figures and/or tables, and approved the final draft.
- Juan Li performed the experiments, analyzed the data, prepared figures and/or tables, and approved the final draft.
- Shangrong Li conceived and designed the experiments, prepared figures and/or tables, authored or reviewed drafts of the article, and approved the final draft.
- Hanbin Xie conceived and designed the experiments, authored or reviewed drafts of the article, and approved the final draft.

### Human Ethics
The following information was supplied relating to ethical approvals (*i.e.*, approving body and any reference numbers):

The study was approved by the Institutional Review Board of the Third Affiliated Hospital of Sun Yat-sen University.

### Clinical Trial Ethics
The following information was supplied relating to ethical approvals (*i.e.*, approving body and any reference numbers):

The Third Affiliated Hospital of Sun Yat-sen University granted Ethical approval to carry out the study within its facilities.

## Field Study Permissions

The following information was supplied relating to field study approvals (*i.e.*, approving body and any reference numbers):

Collect specimens were approved by the Institutional Review Board of the Third Affiliated Hospital of Sun Yat-sen University.

## Data Availability

The raw data are available in the Supplemental Files.

## Clinical Trial Registration

The following information was supplied regarding Clinical Trial registration:

ChiCTR1800019651.

## Supplemental Information

Supplemental information for this article can be found online at http://dx.doi.org/10.7717/peerj.17154#supplemental-information.

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
