# Peer review of "The feasibility of the posterior tibial nerve-flexor hallucis brevis pathway applied in neuromuscular monitoring: a multicentric, controlled, and prospective clinical trial"

_PeerJ, doi:10.7717/peerj.17154_

## Round 0.1 · original submission · Minor Revisions

After a thorough evaluation process, the changes requested by the reviewers were considered to be relatively minor. These changes are intended to enhance the clarity, presentation, and overall quality of your manuscript. Please carefully review the feedback provided by the reviewers and address these revisions in your final manuscript. Once you have completed the minor revisions, please submit the revised manuscript via our online submission system by the deadline. Our editorial team will conduct a final check to verify that all requested revisions have been appropriately addressed.

Reviewer 1 ·

Basic reporting

1.This study aimed to investigate the clinical feasibility of using the flexor hallucis brevis muscle as an alternative site for neuromuscular monitoring, compared to the commonly used adductor pollicis muscle. The study enrolled patients from three different centers who received cis-atracurium, a muscle relaxant commonly used in anesthesia. During the study, two independent monitors were used to monitor the neuromuscular blockade of both the adductor pollicis and the flexor hallucis brevis muscles simultaneously, using a train of four (TOF) pattern. The monitoring continued until the TOF ratios of both sites reached a value greater than 0.9 or until the end of the surgery. The study results showed statistically significant differences between the two monitoring sites in terms of onset time, period of no-twitch response, spontaneous recovery time, and total time. The mean difference for onset time was -53.54 seconds, for the period of no-twitch response was -2.49 minutes, for spontaneous recovery time was 3.22 minutes, and for total time was 5.89 minutes. All these differences were considered statistically significant with a p-value less than 0.001. In conclusion, the use of the posterior tibial nerve-flexor hallucis brevis pathway as a method for monitoring neuromuscular transmission during anesthesia maintenance shows potential as a feasible alternative. However, there are still many areas for improvement in this study.

2.The introduction refers to studies conducted before the knowledge cutoff in September 2021. Without citing more recent studies, it is unclear whether there have been advancements or changes in the understanding of RNMB and its management since then. The reasons for the limited clinical use of quantitative neuromuscular monitoring are not fully addressed in the introduction. Additional information on the challenges or barriers to clinical implementation would provide a more comprehensive understanding of the topic.

3.The writing and structure of the manuscript are in accordance with the standard chapter format. However, The font, format, etc. of the chart in the article should be consistent.

Experimental design

4.neuromuscular monitoring, but it does not sufficiently explain why or provide evidence to support this claim. There is a need for a more thorough comparison of the adductor pollicis and the flexor hallucis brevis regarding their suitability, reliability, and practicality as monitoring sites. The introduction does not clearly state the specific research objectives of the study. It briefly mentions comparing the onset and spontaneous recovery of neuromuscular blockade between the two sites, but it should provide a more explicit statement of the study's aims to guide the reader's understanding.

5. In Methods, the study acknowledges that a drop-out rate of 20% was anticipated. High drop-out rates can impact the validity and reliability of the study results, potentially introducing bias. The study did not mention blinding of the assessors or the patients, which may introduce potential bias in the collection and interpretation of data. In addition, the study primarily focused on intraoperative monitoring and did not include long-term follow-up to evaluate the potential impact or complications associated with the use of the posterior tibial nerve-flexor hallucis brevis pathway.

6. The study mentioned that 213 patients met the inclusion criteria, a total of 154 patients were included in this study and being analyzed. It is indispensable to state what those criteria were to select 154 of 213.

Validity of the findings

7. The study briefly mentions the Bland-Alman plots but does not provide a detailed explanation or interpretation of the patterns observed. A more thorough analysis and discussion of the Bland-Alman plots would enhance the understanding of the agreement between the thumb and toe monitoring sites.

8. The discussion primarily focuses on comparing the results of monitoring at the flexor hallucis brevis and adductor pollicis muscles. While this provides useful information, there is limited discussion on the clinical implications or broader significance of these findings. The discussion could benefit from a more in-depth analysis of the implications for clinical practice and patient outcomes. The discussion mentions previous studies briefly but does not provide a comprehensive discussion or comparison with existing literature. This limits the ability to contextualize the current findings within the broader body of knowledge and to identify any inconsistencies or contradictions with previous research.

9. The discussion does not provide clear directions for future research. Suggestions for further investigation or areas of uncertainty that need to be addressed in future studies would enhance the discussion.

Reviewer 2 ·

Basic reporting

The article by Chen and colleagues provides a well-written and well-designed study that provides valuable insights into the feasibility of using the posterior tibial nerve-flexor hallucis brevis pathway for neuromuscular monitoring during anesthesia. It's well-organized and hits all the essential points, but a few minor things could be improved.

The introduction could be strengthened by providing more information on the limitations of the adductor pollicis and the potential benefits of using the flexor hallucis brevis as an alternative site.

In Figure 3, the font size on the X and Y axes is not readable. The author needs to increase the font size.

Experimental design

No Comment

Validity of the findings

No Comment

Additional comments

A graphical abstract will help summarize the key points and findings of the work.

Reviewer 3 ·

Basic reporting

1.Overall, the manuscript titled "The feasibility of the posterior tibial nerve-flexor hallucis brevis pathway applied in neuromuscular monitoring: a multicentric, controlled, and prospective clinical trial" presents a comprehensive study exploring the potential of using the flexor hallucis brevis as an alternative site for neuromuscular monitoring during anesthesia maintenance. However, there are some instances of awkward phrasing and grammatical errors that need to be addressed for improved clarity and readability. It would be beneficial to have the manuscript proofread by a professional to ensure that the language is consistent and free of errors throughout.
2.The background information of this manuscript needs to be strengthened in order to background the study and its relevance in the field of neuromuscular monitoring. A more comprehensive review of the literature is recommended, especially the last three years of previous studies on alternative sites for neuromuscular monitoring and their findings.
3.Consider restructuring the introduction to flow more smoothly from background information to the specific aims of the study.
4.The article follows a professional structure with clear headings and subsections. Figures and tables are well-designed and effectively complement the text. The Bland-Altman plots and correlation coefficient analysis are particularly informative.

Experimental design

1.The research question is well-defined, relevant, and meaningful. The study aims to compare the onset and recovery of neuromuscular blockade between the adductor pollicis and the flexor hallucis brevis, addressing the need for alternative monitoring sites.
2.Providing a concise summary of the objectives in the introduction section would enhance clarity.
3.Provide more details on the exclusion criteria and dropped patients.
4.Consider breaking down the methods section into subsections for easier navigation and comprehension. Subheadings such as "Study design," "Patient recruitment," "Intraoperative monitoring," "Anesthetic protocol," "Data collection," and "Statistical analysis" can help organize the information more effectively.

Validity of the findings

1.The description of the setup for neuromuscular monitoring using the ulnar nerve-adductor pollicis pathway and the posterior tibial nerve-flexor hallucis brevis pathway is detailed. Consider providing illustrations or diagrams to visually depict the electrode placement and sensor positioning to aid understanding.
2.Provide a comprehensive discussion of the limitations of the study, including the choice of monitoring sites, the use of total intravenous anesthesia, and the exclusion of certain patient populations.

Additional comments

no comment

---

## Round 0.2 · accepted · Accept

After carefully considering the revisions made in response to the reviewers' comments and recommendations, we find that the manuscript now meets the necessary criteria for publication. The reviewers have also recommended accepting the revised version of your manuscript.

Reviewer 1 ·

Basic reporting

no comment

Experimental design

no comment

Validity of the findings

no comment

Additional comments

The authors' revisions and responses generally improved manuscript quality. At the same time, my initial concerns and comments have been addressed, and I think the manuscript is now acceptable.

Reviewer 2 ·

Basic reporting

No Comments

Experimental design

No Comments

Validity of the findings

No Comments

Additional comments

The authors have made changes to my previous comments. I am satisfied with the changes.

Reviewer 3 ·

Basic reporting

The authors have greatly improved the quality of the manuscript through revisions and have largely addressed my concerns.

Experimental design

No comment

Validity of the findings

No comment